# Insights into the Phytochemistry of the Cuban Endemic Medicinal Plant *Phyllanthus orbicularis*: Fideloside, a Novel Bioactive 8-*C*-glycosyl 2,3-Dihydroflavonol

**DOI:** 10.3390/molecules24152855

**Published:** 2019-08-06

**Authors:** Antonio Francioso, Katrin Franke, Claudio Villani, Luciana Mosca, Maria D’Erme, Stefan Frischbutter, Wolfgang Brandt, Angel Sanchez-Lamar, Ludger Wessjohann

**Affiliations:** 1Department of Bioorganic Chemistry, Leibniz Institute of Plant Biochemistry, 06120 Halle (Saale), Germany; 2Department of Biochemical Sciences “A. Rossi Fanelli”, Sapienza University of Rome, 00185 Roma, Italy; 3Department of Plant Biology, Faculty of Biology, University of Havana, 10 200 La Habana, Cuba; 4Department of Chemistry and Technology of Drugs, Sapienza University of Rome, 00185 Roma, Italy; 5Department of Dermatology and Allergy, Charité, Universitätsmedizin Berlin, 10117 Berlin, Germany; 6German Rheumatism Research Centre, a Leibniz Institute, 10117 Berlin, Germany

**Keywords:** *Phyllanthus orbicularis*, *C*-glycoside, flavonoid, natural products, traditional medicine, Cuba, *Phyllanthus chamacristoides*, chromatography, mass spectrometry, NMR, circular dichroism, stereochemistry, Fideloside, cytokines, anti-inflammatory activity

## Abstract

*Phyllanthus orbicularis* (Phyllanthaceae) is an endemic evergreen tropical plant of Cuba that grows in the western part of the island and is used in traditional medicine as an infusion. The aqueous extract of this plant presents a wide range of pharmacological activitiessuch as antimutagenic, antioxidant and antiviral effects. Given the many beneficial effects and the great interest in the development of new pharmacological products from natural sources, the aim of this work was to investigate the phytochemistry of this species and to elucidate the structure of the main bioactive principles. Besides the presence of several known polyphenols, the major constituent was hitherto not described. The chemical structure of this compound, here named Fideloside, was elucidated by means of HR-ESIMS/MS^n^, 1D/2D NMR, FT-IR, and ECD as (2*R*,3*R*)-(−)-3’,4′,5,7-tetrahydroxydihydroflavonol-8-*C*-β-D-glucopyranoside. The compound, as well as the plant aqueous preparations, showed promising bioactive properties, i.e., anti-inflammatory capacity in human explanted monocytes, corroborating future pharmacological use for this new natural C-glycosyl flavanonol.

## 1. Introduction

Natural products represent a very important traditionalsource of novel drugs. They are also a relevant inspiration for the synthesis of novel molecules of pharmaceutical interest. Among the plethora of potential pharmaceutical and nutritional plant-derived molecules, phenolics represent a dominant group of compounds with crucialnatural antioxidantsand flavors [1,2,3,4]. 

With 7500 species of flowering plants, of which 50% are endemic, Cuba hosts more than half of all Caribbean flora [5], that is also the reason why the use of “green” medicine to prevent or treat different illnesses is deeply rooted in Cuban popular traditions. About 1250 species in180 families from Cuba are used as medicinal plants, in which the Euphorbiaceae family is one of the most broadly represented [6,7]. Euphorbiaceae and the segregated Phyllanthraceae are commonly very rich in bioactive metabolites. The genus *Phyllanthus* of this family includes Cuban endemic species which are widely used by traditional medical practitioners for the treatment of different types of diseases [8]. Indeed, other *Phyllanthus* species have worldwide applications including reports from China, the Philippines, Nigeria, East and West Africa, and Latin America comprising further Caribbean countries [9]. Several therapeutic properties have been attributed to this genus, such as antipyretic, antibacterial, antiparasitic, anticontraceptive, and antiviral activities [10,11]. Crude extracts of species such as *Phyllanthus amarus* and *Phyllanthus emblica* have been reported to provide antioxidant and anti-genotoxic activities [12].

*Phyllanthus orbicularis* Kunth is an endemic evergreen plant of Cuba that grows in the western side of the island in Pinar del Rio district. This plant, commonly known as “Alegrìa”, is used in traditional medicine as an infusion for its anti-pyretic and antiviral properties [13,14,15]. In vitro tests showed that the aqueous extract from this species has a marked antiviral activity [16]. In sight of this, several preclinical studies have been carried out with the aim to use this extract as a pharmacological alternative in hepatitis B and human herpes virus type-2 therapy [15,17]. In addition, *P. orbicularis* aqueous extract has anti-mutagenic properties against hydrogen peroxide-induced clastogenicity and mutagenicity exhibiting protective effects against pro-mutagenic aromatics [18,19,20]. Moreover, the extract exhibits a photo-protective activity against γ-radiation, both in pre- and post-irradiation treatments [21,22]. Recent data demonstrated the protective effect of aqueous extracts from Cuban endemic *P. orbicularis* against UV-light induced DNA damage and genotoxicity [23,24,25].

To gain insight into the chemical principles responsible for the biological effects, a complete characterization of the phytochemical profile of Cuban endemic *Phyllanthus* species is required. To date only limited data are available, obtained by GC/MS and HPLC analyses of organic or aqueous extracts, which demonstrate the presence of known bioactive terpenoids and flavonoids [15,26]. *P. orbicularis* aqueous extract is the most interesting preparation from a pharmacological point of view, being the most effective and most studied in biological models, and the most relevant in application. However, despite its beneficial biological activities, the phytochemical composition of this plant is still not completely known. Of all previous works describing *P. orbicularis* aqueous extract chemistry, none offer a complete profile and a detailed chemical characterization. Given its common use in Cuba and the many beneficial effects of the plant, and the still increasing interest in the development of new pharmacological products from natural sources, the aim of this work is to investigate the phytochemistry of the Cuban endemic *P. orbicularis* compared to the endemic *P.chamacristoides*, which is not used in traditional medicine. The focus will be laid on the elucidation of the structures of the still unknown main bioactive principles.

## 2. Results

### 2.1. Phytochemical Characterization of Cuban Phyllanthus Species

UPLC-DAD profiles (280 nm) of aqueous extracts from two endemic Cuban *Phyllanthus* species (*P. orbicularis* and *P. chamacristoides*) analyzed at the same concentration and in the same chromatographic conditionsare shown in Figure 1. We compared the metabolite profile of the medicinal plant *Phyllanthus orbicularis* with that from *Phyllanthus chamacristoides* not used in traditional medicine. The chromatograms can be divided virtually in three regions, representing three different classes of molecules. From the beginning of the chromatographic run to 3.5 min there is the elution of hydrophilic phenolic acids, the central part of the chromatogram (from 3.5 to 4.8 min) is characterized by the presence of catechins and procyanidins (monomers and polymers), and from 4.8 min to the end of the gradient the last eluting molecules of these extracts are represented by more complex flavonoids. Analyzed compounds were numbered from **1** to **9** (Appendix A) in order of their retention times and correspond to the major compounds detected (Table 1). In accordance with previously reported data, our results reveal the presence of catechin (peak **4**, Rt 3.85 min), procyanidin B2 (peak **5**, Rt 4.07 min), epicatechin (peak **6**, Rt 4.23 min) and rutoside (peak **8**, Rt 4.51 min) in *P. orbicularis*. The identity of these compounds was confirmed by the analysis of authentic analytical standards. As Figure 1 shows, peak **3** is the major compound in *P. orbicularis* and is present only in this species. Negative mode HRESIMS analyses of the other eluting peaks evidenced deprotonated ions [M−H]^−^(UVλ_max_) at *m*/*z* 315.0717 (255,sh290 nm), *m*/*z* 355.0668 (326 nm), *m*/*z* 465.1039(290 nm), *m/z* 865.1984 (280 nm), and *m*/*z* 593.1514 (255 and 354 nm) corresponding, respectively, to compounds **1**, **2**, **3**, **7** and **9**. Collected samples were first submitted to direct infusion ESI-MS/MS fragmentation and then the whole extracts were analyzed by LC-HRESI-MS^n^ for further structural elucidations. Molecular fragments obtained by both methods were in accordance with standards and literature data for all compounds investigated. The UV-Vis absorption features of compound **1** (C_13_H_16_O_9_, Rt 3.49 min) fit with the presence of a protocatechuic moiety (255 nm; sh 290 nm). The MS^2^ fragmentation pattern of the parent ion (*m*/*z* 315.0717) is consistent with protocatechuic acid glucoside, showing the presence of the major fragment at *m*/*z* 153.0196 derived from the loss of the sugar moiety. MS^3^ of this fragment gives rise to an ion at *m*/*z* 109 in accordance with the structure of this molecule [27,28]. Compound **2** (C_15_H_16_O_10_, Rt 3.52 min) shows an UV-Vis spectrum with λ_max_ at 326 nm typical of hydroxycinnamate conjugated systems. ESI-MS^2^ fragmentation of the pseudomolecular ion at *m*/*z* 355.0668 generates fragments at *m/z* 147.0301 and *m/z* 163.0405, resulting from two different cleavages of the ester bond, and two complementary fragments at *m*/*z* 209.0304 (C_6_H_9_O_8_^−^) and *m*/*z* 191.0198, referring to glucaric acid and its dehydration product, respectively [29,30,31]. MS^3^ of the fragment at *m*/*z* 191 gives rise to subsequent glucaric acid decarboxylation products at *m/z* 147.1865 and *m/z* 85.0297. Peak **2** was consequently identified as *p*-cumaroyl-glucaric acid. Compounds **7** (Rt 4.35 min) and **9** (Rt 4.6 min) were already detected in *Phyllanthus orbicularis* extracts and were re-identified by means of their chromatographic behavior, spectroscopic features and ESI-HRMS/MS^n^ fragmentation patterns [15]. Compound **7** shows a molecular ion at *m*/*z* 865.1986 and a UV-vis λ_max_ at 281 nm. This molecule was previously identified in this extract as the epicatechin trimer procyanidin C1. The MS/MS fragmentation of the precursor ion generates the dimer at *m*/*z* 577.1353 with the same subsequent MS/MS fragmentation pattern as procyanidin B1/B2 type molecules, confirming the nature of this compound [32]. Peak **9** was identified as the flavonol glycoside nicotiflorin in accordance to the literature data for this plant constituent. MS/MS fragmentation of the pseudomolecular ion (*m*/*z* 593.1514 [M-H]^−^) gives rise to the aglycone part at *m*/*z* 285.0403, corresponding to a kaempferol moiety. Further MS/MS of the fragment at *m*/*z* 285 generates fragments at *m*/*z* 255, 227 and 151 [33].

### 2.2. Structure Elucidation of Compound 3 (Fideloside)

Although compound **3** is the major metabolite of *P. orbicularis* (*m*/*z* 465.1039 [M-H]^−^, Rt 3.65 min), no previous chemical identification was reported [17]. The UV-Vis absorption features of this molecule (λ_max_ at 290 nm) suggested a not completely conjugated flavonoid system and the HRESI-MS derived molecular formula of C_21_H_22_O_12_ indicates the presence of one hexose moiety. MS/MS fragmentation of the precursor ion at *m*/*z* 465.1039 [M-H]^−^ gives rise to a *m*/*z* 345.0620 [(M − H) − 120]^−^ fragment, generated by the cleavage in the glycoside portion on position 2″. These data are consistent with a C-type glycosidic structure [34,35]. Further MS/MSof *m*/*z* 345 generates ions at *m*/*z* 179and 167 resulting from the retro-Diels–Alder fragmentation of flavonoid ring C, indicating the position of the glycoside moiety to be on ring A. Another ion generated by MS^3^ fragmentation of *m*/*z* 345 is the *m*/*z* 277 ion generated by a cleavage in the B ring (Figure 2). 

For completechemical characterization, compound **3** was isolated and the structure was elucidated by means of ^1^H/^13^C 1D and 2D NMR and IR spectroscopy. Table 2 reports NMR data for compound **3**. ^1^H/NMR and ^13^C/NMR spectra in DMSO-d6 display an array of signals in agreement with the hypothesized structure and consistent with literature data of similar flavonoid *C*-glycosides (Appendix A) [35,36,37]. HMBC correlations from the anomeric sugar proton (H-1′’) to C-8, C-7 and C-9 established the presence of the glycosidic moiety at C-8. The diaxial coupling of H-2 and H-3 in the ^1^H NMR spectrum indicates a *trans-*type dihydro-saturation at positions C-2 and C-3. Furthermore, NOESY experiments suggest the positions of phenolic OH groups at C-3′ and C-4′ (Figure 3). The infrared spectrum of compound **3** (Appendix A) is in agreement with the structure, showing representative IR vibrational bands at 3233 cm^−1^ (O–H stretching), 1633 cm^−1^ (C=O stretching), 1362 cm^−1^ (phenolic C–O and O–H vibrational modes), 1277 cm^−1^ (C–O–C stretching in =C–O–C– groups) [38]. 

The absolute configuration was elucidated by comparison of experimental and calculated circular dichroism spectra. The measured spectrum shows the presence of a negative Cotton effect at 295 nm and positive Cotton effect at 331 nm, which correspond to the (2*R*,3*R*) isomer (Figure 4), as previously reported for flavanonols [37,39,40].

The calculated relative conformational energies of the DFT optimized structures are listed in Table 3. The comparison of the Boltzmann weighted calculated CD spectra with the experimental ones clearly indicate that compound **3** adopts a (2*R*,3*R*)-configuration, since the fit with the experimental spectrum for this configuration is much better (Figure 4) than for the (2*S*,3*S*)-configuration (Appendix A).

On the basis of all experimental and calculated data, the chemical structure of this molecule is elucidated as (2*R*,3*R*)-(−)-3’,4’,5,7-tetrahydroxydihydroflavonol-8-*C*-*β*-D-glucopyranoside, a structure never reported previously for the genus *Phyllanthus* or in any other plant. We suggest to name this new natural compound ‘*Fideloside*’ as the start of our work on this Cuban natural product coincided with the death of the long term Cuban leader Fidel Castro Ruz (Birán, 1927–La Habana, 2016).

### 2.3. Modulation of Cytokine Production

Since flavonoids can act in multiple way on inflammatory processes and Fideloside comprises the bioactive aglycon taxifolin, we investigated whether it is able to modulate interleukin production in human monocytes. Figure 5 shows a first biological assessment of pro-inflammatory (IL-1beta, IL-6) and anti-inflammatory (IL-10) cytokine production in human monocytes treated with poly-IC as a pro-inflammatory stimulus. 

These initial results show that the profile of interleukins secretion, in particular IL-10, seems to be modulated by Fideloside similarly to the aqueous extract and aqueous fraction suggesting a crucial role for this compound in the aqueous preparation used in Cuban traditional medicine.

## 3. Discussion

As expected, different polyphenols are the main secondary metabolites in both Cuban *Phyllanthus* species, i.e., *P. orbicularis* and *P. chamacristoides*. *P. orbicularis* presents a more diverse profile of polyphenolic compounds compared to *P. chamacristoides*, and especially a before unidentified dominating compound. Alvarez and co-workers investigated *P. orbicularis* extracts by bioactivity-guided fractionation and determined some phenolic compounds, like catechins and procyanidins, as anti HSV-2 compounds [15]. However, these authors did not characterize the main compound, which together with the identification of several minor constituents, was still lacking [15,17]. Our results demonstrate the presence of phenolic compounds unidentified in previous investigations in the two plant species and, most importantly, of a new C-glycoside flavonoid from *P. orbicularis* which represents the main component (circa 70%) of the aqueous infusion. This compound, and C-glycoside flavonoids in general, were not described for the genus *Phyllanthus* until now. C-glycoside flavonoids are a rare and very interesting class of organic natural compounds, considering that many of them have demonstrated their effectiveness as therapeutic agents such as anti-inflammatory, antioxidant, anticancer and antidiabetic drugs [41]. Compared to common glycosides (*O*- and *N*-), *C*-glycosides present minimal conformational differences with the advantage of being resistant to enzymatic or acidic hydrolysis, since the anomeric center of the acetal group is converted to an ether moiety.

Our finding is highly relevant from both a pharmacological and an analytical point of view, because of its potential as a chemotaxonomic and drug quality analysis tool for the recognition and distinction of effective *P. orbicularis* extracts versus, e.g., closely related species. The other consideration rises up from the evidence that *P. orbicularis* is used in traditional medicine as an aqueous extract whereas *P. chamacristoides* is not. Other than *O*-glycosides, *C-*glycosides are much more stable to hydrolysis during extraction and especially stomach passage (acidic cleavage), as well as in the further absorption and metabolism in humans, i.e., in terms of an increased bioavailability and half-life time in vivo [41]. *C*-linked glycosides, as well as the enzymes involved in their metabolism (*C-*glycosyltranferases and *C*-glycosidases), are extremely rare in nature and absent in human metabolism. These features can confer to the molecule a very unique pharmacokinetic profile and distribution.

Fideloside (**3**) as major flavonoid was tested for its anti-inflammatory capacity on human monocytes and demonstrated an increasing effect on the production of IL-10 anti-inflammatory cytokine with respect to pro-inflammatory mediators. The aqueous extract and the aqueous fraction of *P. orbicularis* show a closely related activity profile, suggesting that the newly discovered natural product could act as the major compound responsible for the pharmacological properties of this medicinal plant. In addition to the biochemical features, its physico-chemical properties are very interesting and promising from a pharmacological point of view. The C-glycosidic moiety strongly increases the water solubility and the bioavailability of the molecule, and, as pointed out above, suggests that its amphiphilic behavior could be longer retained if applied orally. For the same reason, its “green”water extraction is possible not only for direct consumption as in traditional applications, but also should allow for easier processing into standardized formulations. Fideloside is the *C*-8-glycosylated form of taxifolin, a well-known flavanonol found in a wide range of vegetables with strong antioxidant capacity and recently published interesting bioactive properties, including anti-inflammatory activity [42,43,44,45]. To date, the only known natural *C*-glycosylated taxifolin is the *C*-6 glycoside with (2*S*,3*S*) stereochemistry (Ulmoside), which biological activity is still under investigation [37,46]. 

As mentioned before, *C*-linked glycosidic phenols are a very unique class of compounds with future potential in different fields. These modifications of plant secondary metabolites are biochemically and evolutionally not well studied and characterized. Enzymes involved in C-glycosyl bond formation are very little known, or minor amounts of C-glycosides are formed as minor byproduct of O-glycosylation [47]. The latter case is highly unlikely here, thus the elucidation of this new peculiar product from the Cuban endemic *Phyllanthus orbicularis* can represent a starting point for the study and characterization of novel enzymes involved in *C*-glycosyl phenolics biosynthesis.

## 4. Materials and Methods

### 4.1. Chemicals, Reagents and General Procedures

TLC was carried out on silica gel 60 F_254_ plates (Merck, Germany). Spots were visualized under UV light (254 and 365 nm) or by heating after spraying with 2% vanillin solution (in 96%H_2_SO_4_). Low resolution ESI-MS analyses were performed on a SCIEX API-3200 instrument (Applied Biosystems, Concord, Ontario, Canada) combined with a HTC-XT autosampler (CTC Analytics, Zwingen, Switzerland). The samples were introduced via autosampler and 2 µL loop injection. ^1^H, ^13^C NMR and 2D (COSY, HSQC, HMBC and NOESY) spectra were recorded on an Agilent DD2 400 NMR spectrometer and the chemical shifts were referenced to TMS or the solvent residual peak. UV spectra were measured with a Jasco V-560 UV/Vis spectrophotometer. CD spectra were acquired on a Jasco J-815 CD spectrophotometer and the specific rotation was measured with a Jasco P-2000 polarimeter. Infrared spectra (ATR) were recorded using a Thermo Nicolet 5700 FT-IR spectrometer. Rutoside, (+)-catechin and (−)-epicatechin analytical standards were obtained from Sigma Aldrich (Germany). Procyanidin B2 was used from the IPB-NWC in-house reference compound library (isolated from *Bumelia sartorum* Mart.) [48]. All the reagents and solvents used were of analytical and LC-MS grade.

### 4.2. Plant Material

*Phyllanthus orbicularis* Kunth was collected in February 2007 from Cajálbana, Pinar del Río, Cuba. The specimens were authenticated and stored at the Cuban Botanical Garden (No.7/220 HAJB). *Phyllanthus chamaecristoides* was collected in spring 2011 from Guantánamo region, Cuba (20° 28″ 32.7″″ N, 74° 43″ 45.4″″ W). The specimens were authenticated by specialists of Cuban Botanical Garden, and stored in this scientific institution as: *Phyllanthus chamaecristoides* Urbano subsp. *baracoensis* (TB4452). 

### 4.3. Extraction and Compound Isolation

The plant material aerial parts were extracted with two different methods according to literature data affording an aqueous extract (AE) and a crude methanolic extract (CE). 

The aqueous extracts (AEs) were obtained from dried aerial parts (ground leaves and stems) following previously described methods [16,25]. Briefly, dry plant material was extracted in bi-distilled water under continuous shaking (1 g: 7.5 mL) for 4 h at 37 °C, or 2 h at 70 °C. After filtration and lyophilization, the material was stored in a dry, cool place until subsequent analyses. 

Crude extracts (CEs) were obtained by extracting grounddry plant aerial parts with 10 volumes (*w*/*v*) of aqueous methanol (80% MeOH). After an initial 15 min of ultrasonication (bath), the plant material was extracted other two times for 2h with 10 volumes each of 80%MeOH at room temperature without sonication and one last time overnight at room temperature with the same procedure. The combined extracts were filtered and the solvents were evaporated to dryness under reduced pressure using a rotary evaporator. The isolation and purification of compound **3** was performed as follows: 25 g of *Phyllanthus orbilucularis* dry material were extracted with 80% MeOH as described above to obtain CE. The dry extract was resuspended in H_2_O and partitioned by liquid–liquid extraction, first with *n*-hexane and then with ethyl acetate, to give the respective H_2_O fraction (AF, 2.2 gr), *n*-hexane fraction (HF, 129 mg) and EtOAc fraction (EF 159 mg). Before purification, every fraction was dissolved to the same concentration and checked by TLC and UHPLC-DAD-MS. Then, 550 mg of AF were chromatographed on a Sephadex LH-20 (Pharmacia) column using pure MeOH as eluent to obtain 20 mg of **3** (Rf 0.43; silica gel;EtOAc:MeOH:H_2_O/6:1:1)
(*2R,3R*)-(−)-3’,4’,5,7-tetrahydroxydihydroflavonol-8-C-β-d-glucopyranoside (Fideloside):

Yellow amorphous. [α]^25^_D_−10.46 (c 0.5; MeOH). CD (c 0.005; MeOH) [θ]_295_−21,139, [θ]_331_ +5001. UV λ_max_;MeOH: 290 nm. IR data: 3233 cm^−1^, 1633 cm^−1^, 1362 cm^−1^, 1277 cm^−1^.^1^H NMR: (DMSO-d6, 400 MHz) and δ: ^13^C NMR: (DMSO-d6, 101 MHz) see Table 2 HRESIMS [M − H]^−^ calculated for C_21_H_21_O_12_^−^: 465.1038, found 465.1039.

### 4.4. UPLC-DAD-MS and ESI-MS/MS

UPLC-DAD-MS was performed on a Waters Acquity H-Class UPLC system (Waters, Milford, MA, USA), including a quaternary solvent manager (QSM),a sample manager with a flow through needle system (FTN), a photodiode array detector (PDA) anda single-quadruple mass detector with electrospray ionization source (ACQUITY QDa). Chromatography was performed on a Waters C18 HSST3 column (100 mm × 2.1 mm i.d., 1.7 μm particle size). Solvent A was 0.1% aqueous HCOOH and solvent B was 0.1% HCOOH in CH_3_CN. Flow rate was 0.5 mL/min and column temperature was set at 25 °C. Elution was performed isocratically for the first minute with 2% B; from minute 1 to minute 6 solvent B was linearly increased to 55%; from 6 to 10 min 20%A and 80%B; then, in 0.5 min solvent B was set at 100% and maintained for 2 min. The column was re-equilibrated with 98% A and 2%B before the next injection. Samples were dissolved in the mobile phase and 10 μL injected through the needle. The PDA detector was set up in the range 200 to 600nm. Mass spectrometric detection was performed in the negative electrospray ionization mode using nitrogen as nebulizer gas. Analyses were performed in the Total Ion Current (TIC) mode in a mass range 50–1000 *m*/*z*. Capillary voltage was 0.8 kV, cone voltage 30 V, ion source temperature 120°C and probe temperature 600 °C. Direct infusion ESI-MS/MS analyses were performed on UHPLC-DAD collected pure peaks.

### 4.5. LC-High Resolution-MS^n^

Separations were performed with the same chromatographic method described above ona Waters C18 HSST3 column (100 mm × 1 mm i.d., 1.7 μm particle size) using a Dionex Ultimate 3000 UHPLC System, equipped with a quaternary pump, autosampler (100 µL sample loop, partial injection mode, 2 µL injection volume, sample temperature 8 °C), and DAD Detector (Thermo Fisher Scientific, Bremen, Germany). The effluent from the PDA detector was connected on-line to an LTQ-Orbitrap Elite mass spectrometer equipped with a high-temperature electrospray ionization (HESI) ion source, controlled by the Excalibur 2.7 software (Thermo Fisher Scientific, Bremen, Germany) and operated in the negative ion mode. The ion spray voltage was set to 4.0 kV, sheath and auxiliary gases on 20 and 5 psi, respectively. The Orbitrap-MS spectra were acquired within the *m*/*z* range of 50–2000 and resolution of 30,000. The tandem mass spectra were acquired by collision induced dissociation (CID) in linear ion trap (LIT) at 35% normalized collision energy and isolation width of 2.0 *m*/*z*. The fragments were detected at anFT-resolution of 30,000.

### 4.6. Molecular Modeling and DFT Calculations

NMR-measurements of compound **3** clearly indicate a *trans* conformation for C2-C3 hydrogen atoms, which are possible only for a (2*S*,3*S*) or (2*R*,3*R*) configuration even when an axial orientation of the dihydroxybenzyl moiety is taken into consideration. Therefore, CD-spectra where calculated for the two enantiomers and compared with the experimental one. The models were constructed using the Molecular Operating Environment (MOE) software and energy optimized using the MMFF94 molecular mechanics force field [49,50]. Conformational analysis was performed with the low mode conformational search module implemented in MOE. As first result, four conformations for each enantiomer with almost identical energies appeared for the dihydroxybenzyl moiety. The resulting most stable conformation for the sugar moiety was incorporated in an identical fashion for all further calculations. For validation of the obtained results for the entire compounds, corresponding calculations were performed with removed sugar moiety. All four low energy conformations of each enantiomer were optimized by using density functional theory (DFT) with BP86 functional and the def2-TZVPP basis set implemented in the ab initio ORCA 3.0.3 program package [51,52,53,54,55,56]. The influence of the solvent MeOH was included in the DFT calculations using the COSMO model [56]. For the simulation of the CD spectra, the first 50 excited states of each enantiomer were calculated by applying the long-range corrected hybrid functional TD CAM-B3LYP with the def2-TZVP(-f) and def2-TZVP/J basis sets. The CD curves were visualized and compared with the experimental ones with the help of the software SpecDis 1.64 [57].

### 4.7. Human Monocyte Isolation andAssessment of Cytokine Release

PBMCs (peripheral blood mononuclear cells) were isolated from the fresh blood of healthy donors (DRK Berlin, in accordance with the recommendations of the local ethics committee on human studies, Charité, Berlin, Germany) by density gradient centrifugation. Briefly, blood was added to Leucosep tubes (Greiner Bio-one GmbH, Frickenhausen, Germany) containing 15 mL of Leukocyte separation medium (Ficoll-Paque Plus, GE Healthcare Biosciences AB, Uppsala, Sweden) and centrifuged at 840× *g* for 20 min without breaks at room temperature. The interphase (PBMCs) was harvested and washed two times with PBS/BSA/EDTA (PBS + 0.2% BSA + 2 mM EDTA) and centrifuged (20 °C, 15 min, 200× *g*) to remove platelets. Monocytes were isolated using the monocyte isolation Kit II (Miltenyi Biotech GmbH Bergisch Gladbach, Germany) following the manufacturer’s instructions. The purity of monocytes as assessed by flow cytometry was between 90% and 95% (±2%). Cells were seeded at 1 × 105 cells/well in serum free medium (X-Vivo 15, Lonza, Verviers, Belgium) in a round-bottom 96 well plate (Sarstedt AG & Co. KG, Nümbrecht, Germany) in the presence of the extracts or compounds or 0.1% DMSO (as control) and activated with 3 µg/mL poly-IC (Sigma Aldrich, Taufkirchen, Germany). Extracts or compound concentrations used for anti-inflammatory activity assay were previously assessed non-toxic for the cells by flow cytometry analyses. After 24 h of incubation at 37 °C supernatants were harvested and cytokine release was measured using a bead-based multiplex cytokine assay (Cytokine 25-Plex human Procarta Plex Panel 1B, Thermo Fisher Scientific, Darmstadt, Germany) and the Bioplex 200 system (Bio-Rad Laboratories GmbH, München, Germany).

## 5. Conclusions

Our results demonstrate the presence of a new bioactive natural product, (2*R*,3*R*)-(−)-3’,4’,5,7-tetrahydroxydihydroflavonol-8-*C*-*β*-D-glucopyranoside (Fideloside, **3**), here reported for the first time, belonging to the rare but highly-valued class of *C-*glycosylated phenols. The compound was isolated and purified from the endemic Cuban medicinal plant *Phyllanthus orbicularis*, and the chemical structure was unequivocally assessed as C8-glucoside of (2*R*,3*R*)-taxifolin by means of spectrometric, experimental and molecular modeling techniques. While is already taxifolin endowed with antioxidant and anti-inflammatory activities, the *C*-glycosylation could give to this derivative even better pharmacological and pharmaceutical properties. Preliminary experimental results on human monocytes demonstrated a promising bioactivity profile, with a positive modulation of anti-inflammatory mediators with respect to pro-inflammatory ones after cellular pro-inflammatory stimulus. In view of the above, Fideloside and *P. orbicularis* represent an important starting point for further bioactivity studies and of both the development of a standardized phytopharmaceutical product and the biochemistry of *C-*glycosylation. 

## Figures and Tables

**Figure 1 molecules-24-02855-f001:**
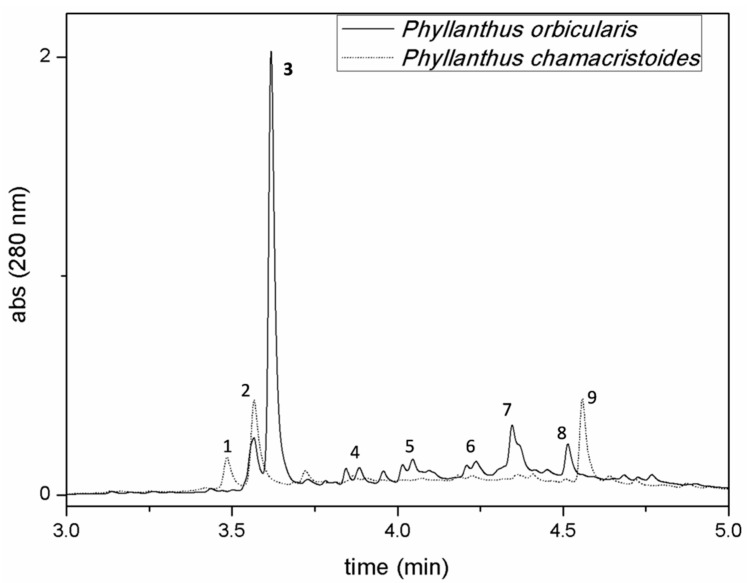
UPLC-DAD (280 nm) chromatograms of *Phyllanthus orbicularis* and *Phyllanthus chamacristoides* aqueous extracts and assignments of eluting peaks.

**Figure 2 molecules-24-02855-f002:**
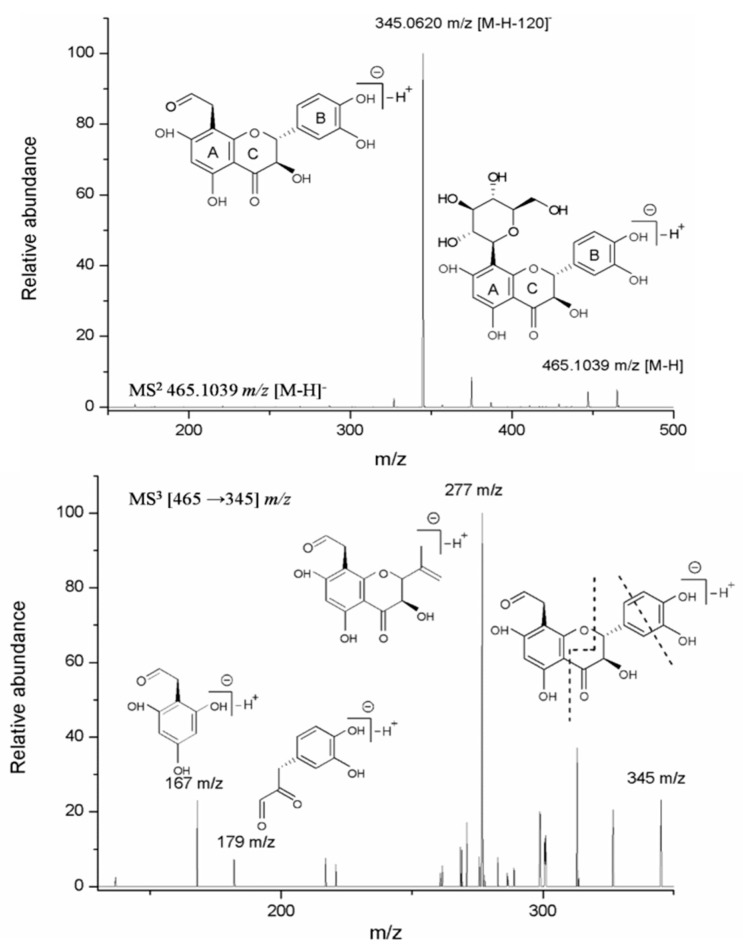
Fideloside (**3**) MS/MS^n^ fragmentation.

**Figure 3 molecules-24-02855-f003:**
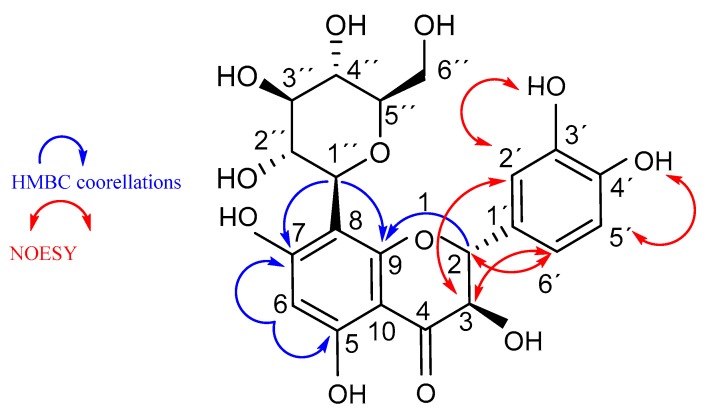
Fideloside (**3**) chemical structure with selected key NMR correlations.

**Figure 4 molecules-24-02855-f004:**
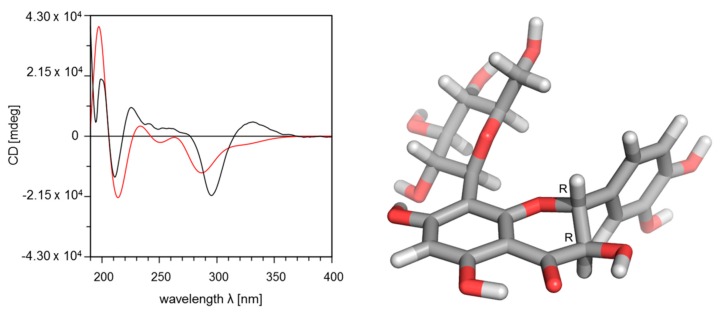
Left: comparison of experimental CD spectrum (black line) with Boltzmann weighted calculated CD spectrum for the (2*R*,3*R*)-enantiomer of compound **3** with a similarity factor S = 0.7099 for sigma = 0.3 eV and 18 nm shift. Right: calculated most stable conformation of the (2*R*,3*R*)-enantiomer.

**Figure 5 molecules-24-02855-f005:**
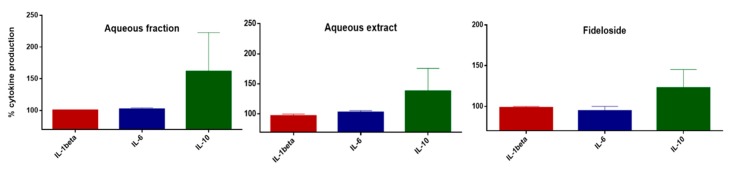
Anti-inflammatory capacity of extracts (3 µg/mL) and isolated Fideloside (1 µM) from *Phyllanthus orbicularis*. Levels of pro- and anti-inflammatory cytokines after poly-IC stimuli (CTRL+, 100%) of human monocytes.

**Table 1 molecules-24-02855-t001:** Spectroscopic and spectrometric data of identified compounds.

Peak	Retention Time (min)	Compound	Molecular Formula	*λ*_max_abs(nm)	MS^1^ [M − H]^−^(*m*/*z*)	MS^2^ [M − H]^−^(*m*/*z*)	MS^3^[M − H]^−^(*m*/*z*)
**1**	3.49	Protocatechuic acid glucoside	C_13_H_16_O_9_	290	315.0717	153.0196	109
**2**	3.52	*p*-Cumaroyl-glucaric acid	C_15_H_16_O_10_	326	355.0668	191.0198	147; 85
**3**	**3.65**	**Fideloside**	**C_21_H_22_O_12_**	**290**	**465.1039**	**345.0620**	**277; 179; 167**
**4** *	3.85	Catechin	C_15_H_14_O_6_	278	289.0720	271.0620; 245.0825	
**5** *	4.07	Procyanidin B2	C_30_H_26_O_12_	280	577.1352	451.1036; 425.088; 289.0720	
**6** *	4.23	Epicatechin	C_15_H_14_O_6_	278	289.0720	271.0620; 245.0825	
**7**	4.35	Procyanidin C1	C_45_H_38_O_18_	281	865.1986	847.1882; 739.1667; 695.1407; 577.1353	[865 → 577] 289
**8** *	4.51	Rutoside	C_27_H_30_O_16_	355	609.1460		
**9**	4.60	Nicotiflorin	C_27_H_30_O_15_	343	593.1514	285.0403	255; 227; 151

* confirmed by analytical standard injection.

**Table 2 molecules-24-02855-t002:** 1D and 2D ^1^H/^13^C NMR data of Fideloside (**3**) in DMSO-d6 as solvent.

Nr.	δ c	DEPT	δ H (J in Hz)	^1^H-^1^H COSY	NOESY	HMBC
2	82.1	CH	5.02 (d, 11.02)	H-3	H-3-; H-6′; H-2′	H-2′;H-6′; OH-3
3	72.1	CH	4.25 (m)	H-2; OH-3	H-2′, H-6′; OH-3	OH-3; H-2
4	197.9	C				H-2; H-6;OH-3
5	162.1	C				H-6; OH-5
6	95.6	CH	6.04 (s)			OH-5
7	165.7	C				H-6; H-1″
8	105.5	C				H-6; H-1″-H; H-2″
9	161.4	C				H-2; H-1″
10	100.5	C				H-6; OH-5
1′	128.4	C				H-2; H-5′; H-2′
2′	115.0	CH	6.94 (brs)	H-6′	H-3; H-2	H-2; H-5′
3′	144.6	C				H-5′
4′	145.1	C				H-2′; H-6′
5′	115.0	CH	6.73 (d, 8.09)	H-6′	H-6′	H-2′; H-6′
6′	118.3	CH	6.84 (brd, 8.09)	H-2′; H-5′	H-2; H-5′; H-3	H-2′; H-2
1″	73.0	CH	4.45 (d, 9.63)	H-2″	H-3″	H-2″; H-6
2″	70.2	CH	3.82 (brt, 9.53)	H-1″; H-3″, OH-2″	H-4″; H-3″; H-1″	H-1″
3″	78.6	CH	3.11 (m)	H-2″; H-4″; OH-3″	H-1″; H-2″	H-1″; H-2″
4″	70.4	CH	2.95 (br)	H-3″; H-5″; OH-4″	H-2″; OH-4; H_2_-6″	H-5″; H-3″; H-1″
5″	81.3	CH	3.09 (m)	H-6″	H-2″; Hb-6″	H-1″
6″	61.7	CH_2_	Ha: 3.70 (m) Hb: 3.43 (m)	H-5″; H-6″; OH-6″	H-6″; H-4″	
3-OH			5.82(d, 6.13)	H-3	H-3	
5-OH			12.01(s)		H-6	
7-OH			-			
3′-OH			8.87 (brs)		H-2′	
4″-OH			9.00 (brs)		H-5′	
2″-OH			4.62 (brs)	H-2″		
3″-OH			4.83 (brs)	H-3″	H-2″, H-4″	
4″-OH			4.84 (brs)	H-4″		
6″-OH			4.57 (brs)	H_2_-6″		

**Table 3 molecules-24-02855-t003:** Results of DFT calculations for the (2*R*,3*R*) enantiomer of compound **3**.

Conformation	O-C2-C1′-C2′ (in°)	C2′-C3′-O-H (in°)	Energy (kcal/mol)	Boltzmann Weight	CD-Fit
1	−61.8	−179.5	0	59.4	0.6757
2	122.8	2.4	0.66	19.5	0.5712
3	−54.7	1.0	0.97	11.5	0.7065
4	121.9	−179.0	1.08	9.6	0.5974
Boltzmann					**0.7099**

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
