# Peer review of "Insights into the Phytochemistry of the Cuban Endemic Medicinal Plant *Phyllanthus orbicularis*: Fideloside, a Novel Bioactive 8-*C*-glycosyl 2,3-Dihydroflavonol"

_molecules, 2019, doi:10.3390/molecules24152855_

Round 1
Reviewer 1 Report
There are many typos in the text as follows.
Line 46: Carribeancountries
Line 56: therapy[15, 17]
Line 58: aromatics[18-20]
Line 78: 280nm
Line 80: conditionsare
Line 85: 4.8min
Line 89: 3.85min
Line 91: fig. 1
Line 93: (255,sh290 nm)
Line 134: 179and
Line 142: 1HNMR and 13CNMR
Line 158 and 227: flavanonols
Line 192: coworkers
Line 195: ofseveralminor constituentwas still
Line 214: glysosyltranferases
Line 241: wascarried
Line 265: methods[16,25]
Line 266: 4h at 37°C,or 2h
Line 270: extractedanother two times for 2h with 10 volumes each of 80%MeOHat
Line 273: 3wasperformed
Line 274: 25g
Line 275: obtainCE
Line 277: 2.2gr), n-hexane fraction (HF, 129mg) and EtOAc fraction (EF 159mg)
Line 280: Rf: 0.43
Line 288: (QSM),a
Line 289: anda
Line 293: from min 1 to min 6
Line 294: 20%A and 80%B
Line 297: 600nm
Line 304: ona
Line 307: 8°C
Line 312: acquiredwithin
Line 317: 3clearly
Line 318: arepossible only for an 2S,3S or 2R,3R
Line 319 and 324: dihydroxyl-benzyl moiety
Line 320: thetwoenantiomers and compared with the experimental one. The modelswere
Line 335: andassessment
Line 348: 0,1%
Line 360: asC8-glucoside
Line 361: andanti-
The following words were recommended to rewrite as follows.
Line 157, 161, 162, 172, 173, 174, 230, 318, 356, 370, and 371: "2R,3R" and “2S,3S” are "(2R,3R)" and “(2S,3S)”, respectively..
Line 203, 210, 214, 222, 234, 238, 358, and 368: “C-glycosyl” is “C-glycosyl”.
Fig. 4 and 6 was not found in the text.
Author Response
Dear reviewer,
We are pleased to send you a fully revised version of our manuscript updated according to your comments and notes (highlighted in yellow).
We thank the referee for giving us the opportunity to improve the quality of our manuscript and better explain our experimental results. We believe that the data are now reported in a much clearer and more exhaustive way. We hope you may find our study now suitable for publication.
Best regards
Dr. Antonio Francioso
Reviewer 2 Report
The paper ‘Insights into the phytochemistry of the Cuban endemic medicinal plant Phyllantus orbicularis: Fideloside, a novel bioactive 8-C-glycosyl 2,3-dihydroflavonol” could be published after minor revision.
In my opinion it is an interesting study, on a topic of interest, because C-glycoside flavonoids are a very interesting class of secondary metabolites in plants.
Some comments:
I advise authors to make careful correction the body of article, because here are a lot of glued words in the article, it is necessary to insert spaces. Same examples: line 40, 43, 60, 80, 85, 89, 99, 128, 134, 195, 241, 266, 270, 273, 275, 277, 278, 289, 297, 301, 304, 317, 318, 320, 325, 335, 360, 361
Line 86 instead ‘identified’ I suggest the word ‘analysed’
Line 90 and 250 instead of the name ‘ rutin’, it is better to use the name ‘rutoside’
If I understand correctly, the authors used DAD detection. In results or in supplementary materials, if possible, please add UV spectra of detected metabolites
Chapter ‘Modulation of cytokine production’ - Since the article is focused on the isolation and identification of secondary metabolites, I ask the authors to consider whether this chapter is unnecessary. Without it, the work will be more consistent. The next publication should include these studies, along with other bioactivity studies,especially antioxidant activities.
Line 230 ‘whose biological is still under investigation’ - Incorrect, please change
Line 242 ‘2% vanillin-H2SO4 solution’- unclear, what is the concentration of sulfuric acid
Author Response
Dear reviewer,
thank you very much for your revision and your comments. We find your suggestion very interesting and relevant to improve the quality of our manuscript and better explain our experimental results.
We believe that data are now reported in a much clearer and more exhaustive way. We hope you may find our study now suitable for publication
All the typos (glued words) were modified and corrected.
Line 86: done
Line 90 and 250: done
Line 230: done
Line 242: corrected with specified concentration
As regards the UV-vis spectra for detected compounds, we believe that these data are not the crucial information for compound structural elucidations and including all the UV-vis spectra would overload the supplementary materials without conferring a real advantage to the reader. Furthermore most of these spectra are available in the literature and are not a discriminating point for structurally related phenols.
Regarding the Chapter "Modulation of cytokine production" we completely agree with your opinion and suggestions and we believe that this preliminary data can serve just as a first biological screening that will be more deepen studied and investigated in a further scientific work including antioxidant and free radical scavenging activities.
We want to thank the reviewer for his/her very useful revision, comments and interesting suggestions.
Your Sincerely
Dr. Antonio Francioso
Reviewer 3 Report
Authors made analyses of aqueous extracts of two Phyllanthus species, as the result of 1.5 min lasting HPLC-DAD analyses they detected 9 compounds. I doubt in that in the case of plant extracts. Is that method was validated? Authors compared the spectra of compounds with their standards. Please add quantitative elaboration of results, that could give information about dominating compounds. Why only analyses of aqueous extracts was made? Or; Why only that was described in the Results? In Materials and Methods is the information that leaves and stems of both species were analyses. Why authors didn’t described and discussed these results. That could give us knowledge about what part of plant is the best as raw material (quantitative analyses is needed). Elucidation of Fideloside is good prepared. And this aspect of work is interesting. Is this compound was known before in other plants? Why is named fideloside? Is it the first detection of this compound in Phyllanthus species? Please propose another tittle of the article; if you studied two Phyllanthus species: P. orbicularis and P. chamacristoides, you made comparative analyses that information have to be given in the title and in the abstract, and in the conclusion. Add more information about P. chamacristoides in the Introduction. Why this species was studied too? Describe the differences. l. 65 – provide names of identified compounds. Please give more information about known before your studies chemical composition of studied plants I noted many editorial errors: e.g.: P.orbicularis should be P. orbicularis; l.4 novel or new?; l. 68 – compositionof; l. 40- in180; l.124 - Phyllanthuschamacristoides - all should be written separately.Author Response
Dear reviewer,
thanks for your comments, suggestions and interesting notes (elucidations highlighted in green in the revised text)
Regarding the methodological questions we want to point out that we used UPLC (line 123, Fig 1) and not HPLC as an analytical chromatographic method. I think probably this methodological detail was missed. With ultra-pressure chromatography the reported results and time of analysis are perfectly in accordance with the "state of the art".
We also want to point out that we are not validating a method for the analytical quantification of phenols. We are using UPLC coupled with DAD or MS or MS/MS (or column purification in case of preparative scale) to characterize the known and unknown compounds. Moreover, in Fig.1 the relative area of the two chromatograms are comparable (in terms of dominating phenols) being the aqueous extracts from the two endemic Cuban Phyllanthusspecies (P. orbicularis and P. chamacristoides) analyzed at the same concentration and in the same chromatographic conditions. Another major point to remark (highlighted in green in the text, line 197) is that we used the aqueous extracts because all the work is based on the identification of the bioactive principles of Phyllanthus medicinal plant, which is used in Cuban traditional medicine as an aqueous infusion/extract (AE).
As already discussed in the text we also prepared an aqueous fraction starting from the crude extract and we compared the differences in biological activities with respect to pure major purified compound (Fideloside).
As regards the name of the novel discovered compound the explanation of the genesis of the name Fideloside is already given in the text (now highlighted in green, line 164). In the same paragraph we already underlined that this compound is here detected and structurally characterized for the first time (not only in the genus Phyllanthus, but also in any other plant or organism).
In lines 72 and 80 (highlighted in green) we better stressed why we used also P. chaemacristoides. Basically we analyzed this plant extract in the same analytical conditions to have a controlled endemic Cuban Phyllanthus species which is not traditionally used in popular Cuban medicine.
All the typos and editorial errors were corrected as request.
Sincerely
Dr Antonio Francioso
Round 2
Reviewer 3 Report
All my comments were clarified and applied.